# Vertical Oscillation of Railway Vehicle Chassis with Asymmetry Effect Consideration

**DOI:** 10.3390/s22114033

**Published:** 2022-05-26

**Authors:** Frantisek Klimenda, Jan Skocilas, Blanka Skocilasova, Josef Soukup, Roman Cizek

**Affiliations:** 1Faculty of Mechanical Engineering, University of Jan Evangelista Purkyne in Usti nad Labem, Pasteurova 1, 400 96 Usti nad Labem, Czech Republic; blanka.skocilasova@ujep.cz (B.S.); josef.soukup@ujep.cz (J.S.); roman.cizek@ujep.cz (R.C.); 2Faculty of Mechanical Engineering, Czech Technical University in Prague, 4 Technicka, 166 07 Prague, Czech Republic; jan.skocilas@fs.cvut.cz

**Keywords:** vehicle model, symmetry and asymmetry loading, kinematic excitation, motion equations, experimental results

## Abstract

This paper deals with the problem of vertical oscillation of rail and road vehicles under symmetrical and asymmetrical loading and symmetrical and asymmetrical kinematic excitation. The term asymmetry is understood as the asymmetric distribution of vehicle mass and elastic and dissipative elements with respect to the axes of geometric symmetry, including asymmetric kinematic excitation. The various models used (spatial, planar, quarter-plane) are discussed and their analytical solutions are outlined. The theory of the spatial model is applied to the chassis of a model railway vehicle. The basic relations for the calculation of the equations of motion of this vehicle are given. In the next section, the experimental solution of a four-axle platform rail car is described and the measurements of vertical displacement and accelerations when crossing wedges (representing unevenness) are given.

## 1. Introduction

The problem of vibration in wheeled vehicles, both road and rail, is solved based on different assumptions and models, under different operating conditions. There is extensive literature on this issue, e.g., [1,2,3,4,5,6,7]. The solutions are represented by numerical methods [8] considering bogie track interaction with a system of nine degrees of freedom. The track unevenness excitation of vibration has been investigated by several authors [9] using a 68-degrees of freedom model, where results are compared with experiments. Various frequency ranges of dynamic system behaviour were investigated in [10]. One review summarises actual state of art of the results of railway vehicle vibration induced by dynamic impact loadings [11]. Furthermore, interaction between the wheel and the track has significant impact on the ride safety and control, resulting in the wearing of the wheel. A review of this problem is presented in another study [12]. The wheel flat defect caused by the above-mentioned phenomena and its impact on the wheel/rail dynamics can be described by a simple 3D model [13]. The excitation of the mechanical system (railway car) oscillation is usually modelled by passing the wheel over the uneven section of the road. Other types of irregularities such as stiffness irregularities, irregularities from different track positions, and irregularities in the wave propagation are investigated in [14] and also have an import role in drive comfort and safety. Other authors also paid attention to this issue from an experimental work perspective [15]. 

An overview of the solution to the vibrations of a spatially elastic body with consideration of various influences are given, for example, in [16]. The suppression of the influence of inadmissible vibrations and shocks in the investigation of the vibrations of suspended parts of rail and road vehicles is dealt with, for example, in [17,18,19,20]. The initial problem lies in investigating the vibrations of more complex systems of rigid bodies by elastic couplings. The most commonly used model for investigating the vibrations of a real structure is a system model with one or more degrees of freedom, solved as a one-dimensional chain of material points moving along a single line (or along straight parallel lines), possibly with the consideration of energy dissipation [21].

A major disadvantage of the current models, i.e., most often quarter-plane or planar, but also spatial, is the neglect of the effect of asymmetry on their driving characteristics. Therefore, only solutions to partial problems are given, mostly for half-plane (planar) models, which assume planar symmetry with one longitudinal axis of symmetry, respect the effect of the displacement of the centre of gravity from the centre of geometry, or consider different intensities of viscous damping and their combination. However, the most common solutions to the oscillation of vehicle models are based on quaternion models, usually vertical and elastic coupling with dissipative elements, and with different numbers of bodies—these are models with multiple degrees of freedom with coupled vertical displacements. In the application of the quaternary model, two axes of vehicle symmetry are assumed, i.e., full symmetry and its kinematic excitation.

It is known from practice that vehicle crashes, both road and rail, often occur for reasons that are difficult to explain. In the case of road vehicles, the cause is often cited as not adapting the speed to the road conditions, etc. In the case of railway vehicles, the causes are often more difficult to determine, apart from the obvious causes such as rail breakage, rail gauge collapse, etc. The cause is often due to incorrect load distribution and the resulting asymmetry of the load on the individual axles or wheels. In the case of vehicle oscillation, vertical vibrations can cause loss of wheel contact with the road (loss of vehicle controllability), and in the case of railway vehicles, loss of wheel contact with the rail due to unevenness of the rail, resulting in derailment.

Therefore, the investigation of the effect of asymmetry on the vertical oscillation of the model vehicle must be carried out by analysing the various causes and their consequences on spatial models.

When analysing the effect of asymmetry on the vertical oscillation of vehicles, we must distinguish three basic cases of the effect of asymmetry with respect to the axes of geometric symmetry, which are determined by two mutually perpendicular axes, the symmetry of the vehicle’s track and wheelbase, and the intersection at the geometric centre of the vehicle:Asymmetry of the vehicle weight distribution with respect to the axes of geometric symmetry, the position of the centre of gravity, the directions of the main central axes of inertia, both of the vehicle structure itself and of the loaded vehicleAsymmetry of the geometry of the distribution of elastic and dissipative elements of the couplings of the individual bodies of the vehicle system, their mechanical properties, spring stiffnesses, intensity of viscous damping assuming linear couplings of the individual quantities, and small displacements and rotations of the parts of the systemAsymmetry of kinematic excitation, i.e., the field of unevenness of the road or track surface, which define the kinematic excitation of the system at the point of contact between wheel-vehicle or wheel-track.

The above types of asymmetry can exist separately or together, and the third case is the most common.

The basic prerequisite for investigating the influence of asymmetry on the vibration of a system of bodies (a vehicle) is the choice of a suitable simple spatial model that allows for the investigation of different cases of asymmetry and the comparison or verification of the results obtained by different methods, i.e., theoretical (analytical, numerical, and simulation) and experimental. Most often, we compare the vertical displacement of the centre of gravity and the rotation about the axes passing through the centre of gravity and the vertical movements of any point of the vehicle.

In this study, a methodology for solving the vertical vibration of a vehicle was developed and this methodology was applied to several real vehicle cases (wheeled and tracked) to verify its applicability to a large set of vehicles. The methodology also allows for the determination of the magnitude and time history of wheel forces, the deformation magnitude or allowability of the individual elements of the spring links and parts of the structures, and the damping intensity criteria, including the fulfilment of the physiological requirements for the vehicle ride.

## 2. Theoretical Solution

We use the basic theory of motion equations of an elastically supported rigid body when solving the vertical vibration of the vehicles, both road and rail. A rigid body of general shape can be elastically supported or suspended in an inertial orthonormal Cartesian space, which is defined by a stationary component system. In our case, it is a more complex system where we solve the vertical oscillation of the vehicle (model) including the effect of asymmetry based on various assumptions. As already mentioned, a quarter, half, or full 3D model of the mechanical system with several degrees of freedom, with different mass distributions and with different elastic and dissipative elements (bindings) is used. The kinetic excitation is considered, and the solution will be demonstrated analytically, numerically, and experimentally.

For the basic analytical derivation, a simple but sufficiently general 3D model was designed to meet the requirements outlined in the previous section. The proposed model (Figure 1) is suitable for both the analytical derivation of relationships and experimental verification. 

The model consists of a rigid plate, which is supported on four springs (*k*_1_–*k*_4_). The springs are supplemented by dampers (*b*_1_–*b*_4_). Weights *z*_1_ and *z*_2_ are placed on the plate to simulate the imbalance of the vehicle. The centre of gravity of the vehicle *T* is displaced in the *x*-axis direction by *e_x_* and in the *y*-axis direction by *e_y_* relative to the geometric centre *C*. The *x* and *y* axes passing through the geometric are the axle base and wheel spacing axes. The rotation of the model about the *x*-axis is *φ_x_* and about the *y*-axis is *φ_y_*. The value of *h* is the height of the road unevenness.

The time course of vertical displacement, i.e., the change in the position of the centre of gravity *w_T_*(*t*) and the rotation of the model about the central axes of inertia *φ_x_*(*t*), *φ_y_*(*t*), was chosen as the criterion for comparing the individual cases.

In general, we can describe the motion of any point A (*x*_A_, *y*_A_) in terms of its vertical displacement, velocity, and acceleration versus time (e.g. at the suspension location) and determine the wheel pressures of the vehicle (Equations (1)–(3)).
(1)$$w_{A}\left(t\right){=w}_{T}\left(t\right){+x}_{A\;}\varphi_{y}\left(t\right)-{\;y}_{A\;}\varphi_{x}\left(t\right),$$
(2)$${\dot{w}}_{A}\left(t\right)={\dot{w}}_{T}\left(t\right)+x_{A}{\dot{\varphi }}_{y}\left(t\right)-y_{A}{\dot{\varphi }}_{x}\left(t\right),$$
(3)$${\ddot{w}}_{A}\left(t\right)={\ddot{w}}_{T}\left(t\right){+x}_{A\;}{\ddot{\varphi }}_{y}\left(t\right)-{\;y}_{A\;}{\ddot{\varphi }}_{x}\left(t\right).$$

### 2.1. Individual Models Motion Equations 

The solution was carried out for a spatial model (3D), a planar model, and a quartic model. The solution is found by determining the equations of motion of a mechanical system with three degrees of freedom.

#### 2.1.1. Spatial Model

The equations of motion for the vibration of a spatial model of an elastically supported rigid body were determined from the Lagrange equations of the second kind
(4)$$\frac{d}{dt}\left(\frac{\partial E_{k}}{\partial {\dot{q}}_{j}}\right)-\frac{\partial E_{k}}{\partial q_{j}}+\frac{\partial E_{p}}{\partial q_{j}}+\frac{\partial R_{d}}{\partial {\dot{q}}_{j}}=Q_{j}\;\;\;\;\;\mathrm{for}\;j=1,2,\ldots p,$$
where *p*—number of degrees of freedom, *q_j_*—generalised coordinate, *E_k_*—kinetic energy, *E_p_*—potential energy, *R_d_*—Rayleigh dissipated energy, *Q_j_*—excitation forces.

The solution procedure is known from the literature. After substituting for the individual variables and performing the appropriate derivations, we obtain a system of linear inhomogeneous equations, which can be written in matrix form
(5)$$M_{h}{\ddot{q}}_{j}+M_{b}{\dot{q}}_{j}+M_{k}q_{j}=Q_{j}\left(q_{j^{\prime }}{\dot{q}}_{j^{\prime }}t\right)\;\;\;\;\;\mathrm{for}\;j=1,2,3,$$
where **M***_h_*—mass matrix, **M***_b_*—damping matrix, **M***_k_*—stiffness matrix, **Q***_j_*—vector of generalised excitation function (in our case kinematic excitation of the type $Q=k_{j}q_{j}+b_{j}{\dot{q}}_{j}$).

To solve Equation (5), we use Laplace integral transforms, modifying the elements of the matrix **M***_h_* by multiplying it from the left by the diagonal matrix **D** = (*d_ij_*) of the third order, whose elements are equal to the inverse of the diagonal elements of the matrix **M***_h_*.

In the case of a symmetrical mechanical system, the mass matrix is unitary. If we multiply Equation (5) by the diagonal matrix **D** from the left, we get
(6)$${\mathrm{DM}}_{h}{\ddot{q}}_{j}\;+\;{\mathrm{DM}}_{b}{\dot{q}}_{j}\;+\;{\mathrm{DM}}_{k}q_{j}\;=\;{\mathrm{DQ}}_{j}.$$


We rewrite this equation in the form
(7)$$M{\ddot{q}}_{j}(t)\;+\;B{\dot{q}}_{j}(t)\;+\;{Kq}_{j}({t)\;=\;F}_{j}\left(t\right),$$
where **M**—mass matrix, **B**—damping matrix, **K**—stiffness matrix, **F***_j_*—vector of generalised kinematic excitation function.

The solution to Equation (7) is found by determining the elements of the damping matrix *b_ij_* and the stiffness matrix *a_ij_*. The calculation of these elements is given, e.g., in [22].

If the vertical component of the force at the *m*-th wheel location between the wheel and the ground is given by
(8)$$\Phi_{m}\left(t\right){\;=\;k}_{m}h_{m}\left(t\right){+\;b}_{m}{\dot{h}}_{m}\left(t\right),$$
where *k_m_*—spring stiffness of *m*-th wheel, *h_m_*(*t*)—the height of the unevenness at the point of contact of *m*-th wheel, *b_m_*—damping coefficient of *m*-th wheel, then the component of the generalised function *F*_1_(*t*)—is physically the acceleration component of the vertical displacement [ms^−2^] and is given by the ratio of the product of the vertical components of the wheel forces and the mass of the vehicle model
(9)$$F_{1}\left(t\right){=m}^{-1}\overset{4}{\underset{m=1}{\sum }}\Phi_{m}\left(t\right).$$

The component of the generalised function *F*_2_(*t*)—the physical component of the angular acceleration with respect to the *x*-axis [s^−2^]—is given by the ratio of the product of the moments of the components of the wheel forces to the *x*-axis and the moment of inertia of the vehicle model to this axis
(10)$$F_{2}\left(t\right){=J}_{x}^{-1}\left[-\overset{2}{\underset{m=1}{\sum }}\Phi_{m}\left(t\right)l_{ym}+\overset{4}{\underset{m=3}{\sum }}\Phi_{m}\left(t\right)l_{ym}{+e}_{y}\overset{4}{\underset{m=1}{\sum }}\Phi_{m}\left(t\right)\right].$$

The component of the generalised function *F*_3_(*t*)—physically the component of the angular acceleration with respect to the *y*-axis [s^−2^]—is given by the ratio of the product of the moments of the wheel force components to the *y*-axis and the moment of inertia of the vehicle model to this axis
(11)$$F_{3}\left(t\right){=J}_{y}^{-1}\left[\overset{}{\underset{m=1,4}{\sum }}\Phi_{m}\left(t\right)l_{xm}+\overset{}{\underset{m=2,3}{\sum }}\Phi_{m}\left(t\right)l_{xm}{+e}_{x}\overset{4}{\underset{m=1}{\sum }}\Phi_{m}\left(t\right)\right],\;$$
where *l_xm_* a *l_ym_* for *m* = 1, 2, 3, 4 are the distances of the support (elastic viscoelastic damping) from the axes $\bar{x}$ and $\bar{y}$ of the geometric symmetry with the origin at point *C*, *J_x_*, *J_y_*—moments of inertia of the vehicle mass to the respective axes *l_xm_*, *l_ym_*.

By defining the elements of the matrices **M**, **B**, **K** and the components of the vector **F**, a system of simultaneous linear inhomogeneous differential equations, the equations of motion of a spatial model of a vehicle with three degrees of freedom, with complete asymmetry: the mass distribution of the system ($e_{x}\;\neq {\;0,\;e}_{y}\;\neq {\;0,\;D}_{xy}\;\neq {\;0,\;D}_{yx}\;\neq \;0$) of the geometry and stiffness of the elastic support, the geometry and intensity of the viscous damping, and with the asymmetry of the kinematic excitation defined by the road roughness *h*(*x*) → *h*(*t*) is determined.

It should be emphasised that the investigation of vehicle oscillations assuming a planar vehicle model with a longitudinal axis of symmetry, and especially assuming a quarter vehicle model with both longitudinal and transverse axes of symmetry, will substantially change the definition of the elements of the matrices of the equations of motion (7) and thus the required solution. This can be documented by comparing the matrix elements of the single-individual vehicle models.

#### 2.1.2. Planar Model

For spatial mode of vehicle with a longitudinal axis of symmetry, which has only two degrees of freedom, *w*(*t*) and *φ_y_*(*t*), the following can be assumed:Mass matrix **M** is diagonal and unit, and for mass distribution is valid *D_xy_* = *D_yx_* = 0, => *S*_23_ = *S*_32_ = 0, *e_y_* = 0.The geometry of support is defined by dimensions *l_y_*_1_ = *l_y_*_4_, *l_y_*_2_ = *l_y_*_3_, *l_x_*_1_ = *l_x_*_4_, *l_x_*_2_ = *l_x_*_3._Damping intensity *b*_1_ = *b*_4_, *b*_2_ = *b*_3_.Stiffness of elastic support is given by formulas *k*_1_ = *k*_4_, *k*_2_ = *k*_3_.

Then damping matrix **B** elements are
(12)$$\begin{array}{ccc} b_{11}{=2m}^{-1}\left(b_{1}{+b}_{2}\right), & b_{12}=0, & b_{13}{=2m}^{-1}\left(b_{1}l_{x1}-{\;b}_{2}l_{x2}\right)-{\;e}_{x}b_{11},\\ b_{21}=0, & b_{22}{=J}_{x}^{-1}2\overset{3}{\underset{j=1}{\sum }}b_{j}l_{yj}^{2} & b_{23}=0,\\ b_{31}=mJ_{x}^{-1}b_{13}, & b_{32}=0, & b_{33}{=J}_{y}^{-1}\left[2\sum_{j=1}^{2}b_{j}l_{xj}^{2}{+e}_{x}^{2}b_{11}m\right]. \end{array}\;$$

Stiffness matrix **K** elements are
(13)$$\begin{array}{ccc} a_{11}{=2m}^{-1}\left(k_{1}{+k}_{2}\right), & a_{12}=0, & a_{13}{=2m}^{-1}\left(k_{1}l_{x1}-k_{2}l_{x2}\right)-{\;e}_{x}a_{11},\\ a_{21}=0, & a_{22}{=2J}_{x}^{-1}\sum_{j=1}^{2}k_{j}l_{yj}^{2}, & a_{23}=0,\\ a_{31}=mJ_{x}^{-1}a_{13}, & a_{32}=0, & a_{33}{=J}_{y}^{-1}\left[2\sum_{j=1}^{2}k_{j}l_{xj}^{2}{+e}_{x}^{2}a_{11}m\right]. \end{array}\;$$

Based on the above relations, we can determine the components of the vertical forces acting at the location of the *m*-th wheel between the wheel and the road (rail) for crossing over unevenness of height *h*_1_ = *h*_4_, *h*_2_ = *h*_3_
(14)$$\begin{array}{c} F_{1}{\;=\;2m}^{-1}\left[\sum_{j\;=\;1}^{2}k_{j}h_{j}\left(t\right)\;+\;\sum_{j\;=\;1}^{2}b_{j}{\dot{h}}_{j}\left(t\right)\right]{,\;\;\;\;\;\;F}_{2}\;=\;0,\\ F_{3}\left(t\right){\;=\;J}_{y}^{-1}\left[\sum_{m\;=\;1,4}^{}\Phi_{m}\left(t\right)l_{xm}\;-\;\sum_{m\;=\;2,3}^{}\Phi_{m}\left(t\right)l_{\mathrm{xm}}{\;+\;e}_{x}\sum_{m\;=\;1}^{4}\Phi_{m}\left(t\right)\right]\; \end{array}$$


#### 2.1.3. Quarter Model

For quarter model of vehicle with a longitudinal and transverse axis of symmetry, i.e., for a system of one degree of freedom *w*(*t*) the following is assumed:Mass distribution *D_xy_* = *D_yx_* = 0 → *S*_23_ = *S*_32_ = 0, *e_x_* = 0, *e_y_* = 0.Mass matric **M** is unit, the centre of gravity of the system is identical to the centre of geometric gravity *C*, the main central axes of inertia are identical to axes of geometrical symmetry.The support geometry is determined by dimensions *l_xj_ = l_x_, l_yj_ = l_y_* for *j* = 1, 2, 3, 4.Damping intensity *_j_* = *b*, for *j* = 1, 2, 3, 4.Stiffness of elastic support is given by *k_j_ = k*.

Then, damping matrix **B** elements are
(15)$$\begin{array}{ccc} b_{11}{=4m}^{-1}b,\; & b_{12\;}=0, & b_{13\;}=0,\\ b_{21}=0, & b_{22}{=4J}_{x}^{-1}{kl}_{y}^{2}, & b_{23}=0,\\ b_{31}=0, & b_{32\;}=0, & b_{33}{=4J}_{y}^{-1}{kl}_{x}^{2}. \end{array}\;$$

Stiffness matrix **K** elements are
(16)$$\begin{array}{ccc} a_{11}{=4m}^{-1}k, & a_{12}=0, & a_{13}=0,\\ a_{21}=0, & a_{22\;}{=4J}_{x}^{-1}{kl}_{y}^{2}, & a_{23}=0,\\ a_{31}=0, & a_{32}=0, & a_{33}{=4J}_{y}^{-1}{kl}_{x}^{2}. \end{array}\;$$

Components of the generalised kinematic excitation function vector for *k_j_* = *h*
(17)$$\begin{array}{ccc} F_{1}{\;=\;4m}^{-1}\left[kh\left(t\right)\;+\;b\dot{h}\left(t\right)\right], & F_{2}\;=\;0, & F_{3}\;=\;0. \end{array}\;$$


#### 2.1.4. Models Discussion

From the brief analysis of the equations of motion discussed above, and the forces on the individual wheels, it follows that both the plane and quarter models require complete symmetry along the longitudinal axis (plane model) or along both the longitudinal and transverse axes (quarter model).

Due to the distribution of the different structural groups in the vehicle, no vehicle is geometrically symmetrical, it is clear that neither the quarter nor the half model are able to determine the dynamic properties of the vehicle with sufficient accuracy. In addition, the asymmetry in the loading of the vehicle (both cargo and possibly passengers) must be taken into account. For this reason, it is advisable to use a full spatial model for the further solution, which may be generally unsymmetrical (inertia and geometric axes are not parallel) or only unsymmetrical but with parallel inertia and geometric symmetry axes.

### 2.2. Simple Model of Four Axles Wagon

We will show the solution of this system on the chassis of a model of a railway vehicle, considering the effect of asymmetry. We assume a forced oscillation of the model, which is represented by a spatially elastically supported rigid plate with single and multiple primary linear suspension by coil springs (two-axle railway vehicle model). It is a kinematically excited system of three rigid bodies elastically supported and bounded, considering the effect of asymmetry [1,23,24]. However, these authors did not address the issue of asymmetry.

A simple computational model of the vehicle chassis was chosen for the calculation. The model consists of two two-axle chassis with simple primary suspension and simple vehicle body suspension. The chassis are replaced by rigid plates of mass *m*_1_ and *m*_2_ with symmetric mass distribution (centre of gravity is identical to the kinematic centre, and the main central axes of inertia are identical to the axes of geometric symmetry). However, we consider the asymmetry of the spring stiffness parameter and their geometrical support. The model of the vehicle body is considered as a rigid plate of mass *m* for which an asymmetric mass distribution is considered (the position of the centre of gravity is deflected by *e_x_* and *e_y_* from the geometric centre, and the main central axes of inertia are rotated with respect to the axes of geometric symmetry). The computational model is shown in Figure 2.

The derivation of the following relationships is based on general assumptions, i.e., body stiffness, small displacements and rotations, and linear spring characteristics. We consider only the vertical displacements of arbitrary points of individual bodies. This vertical change in the position of a point of a body is determined by the displacement of the body’s centre of gravity *w*, *w*_1_ and *w*_2_, the rotations *φ**_x_*, *φ*_*x*1_, *φ*_*x*2_, *φ_y_*, *φ*_*y*1_, *φ*_*y*2_ about the central axes of inertia of individual masses *m*, *m*_1_, *m*_2_ and its distance from these axes. Thus, we solve a system of bodies with nine degrees of freedom.

The equations of motion are derived from the Lagrange equations of the second kind (4). Kinetic energy of the system
(18)$$\begin{array}{c} E_{k}=\frac{1}{2}m{\dot{w}}^{2}+\frac{1}{2}\left(J_{x}{\dot{\varphi }}_{x}^{2}+J_{y}{\dot{\varphi }}_{y}^{2}\;-\;2D_{xy}{\dot{\varphi }}_{x}{\dot{\varphi }}_{y}\right)+\frac{1}{2}m_{1}{\dot{w}}_{1}^{2}+\frac{1}{2}J_{x1}{\dot{\varphi }}_{x1}^{2}+\frac{1}{2}J_{y1}{\dot{\varphi }}_{y1}^{2}+\\ +\frac{1}{2}m_{2}{\dot{w}}_{2}^{2}+\frac{1}{2}J_{x2}{\dot{\varphi }}_{x2}^{2}+\frac{1}{2}J_{y2}{\dot{\varphi }}_{y2}^{2}, \end{array}\;$$
where *J_x_*, *J_y_*—moments of inertia, *D_xy_*—deviation moment to central axes of inertia of chassis with mass *m*, *J_x_*_1_, *J_y_*_1_, *J_x_*_2_ and *J_y_*_2_—inertia moments to the main central axes of bogies with mass *m*_1_, *m*_2_.

The potential energy of the system depends on the displacements of the individual masses in the places of their elastic support, i.e., in the places of their elastic bound. In our case, in the chassis model we have chosen the system of marking for points *A_jki_*, their coordinates *x_jki_*, *y_jki_*, vertical displacements *w_jki_* and stiffness constants *k_jki_*. The individual indices correspond to masses *j* = 1, 2, quadrants *k* = 1, 2, 3, 4 and spring order *i* = 1, 2, ...., *n* (for the case of multiple support).

Similarly, the vertical change *h_jki_* is marked, which is at the point of contact of the axle (spring) with the rigid base, to which we relate the position of the body. This vertical change represents crossing over an unevenness in the road (rail)—kinematic excitation.

We can therefore determine the displacements of the individual points of the chassis, which are given by the relations for body *m*_1_, *j* = 1
*Point**Vertical displacement**Stiffness constant*
*A*_111_$w_{111}{=w}_{1\;}-y_{111}\varphi_{x1}{+x}_{111}\varphi_{y1}-{\;h}_{111,}$$k_{111}$(19)*A*_121_$w_{121}{=w}_{1\;}-{\;y}_{121}\varphi_{x1}-{\;x}_{121}\varphi_{y1}-{\;h}_{121},$$k_{121}$*A*_131_$w_{131}{=w}_{1}{+y}_{131}\varphi_{x1}-{\;x}_{131}\varphi_{y1}-{\;h}_{131,}$$k_{131}$*A*_141_$w_{141}{=w}_{1}{+y}_{141}\varphi_{x1}{+x}_{141}\varphi_{y1}-{\;h}_{141.}$$k_{141}$

For body *m*_2_, *j* = 2
*Point**Vertical displacement**Stiffness constant*
*A*_211_$w_{211}{=w}_{2}{+y}_{211}\varphi_{x2}{+x}_{211}\varphi_{y2}-{\;h}_{211,}$$k_{211}$(20)*A*_221_$w_{221}{=w}_{2}-{\;y}_{221}\varphi_{x2}-{\;x}_{221}\varphi_{y2}-{\;h}_{221},$$k_{221}$*A*_231_$w_{231}{=w}_{2}{+y}_{231}\varphi_{x2}-{\;x}_{231}\varphi_{y2}-{\;h}_{231,}$$k_{231}$*A*_241_$w_{241}{=w}_{2}{+y}_{241}\varphi_{x2}{+x}_{241}\varphi_{y2}-{\;h}_{241.}$$k_{241}$

In the case of a model of a body with a simple spring suspension in the axis of the chassis, the division into quadrants cannot be used to mark the individual points of action of the springs. Marking of the points, coordinates, and stiffness constants *B_jki_*, *x_jki_*, *y_jki_*, *k_jki_*, *j* = 0 body, mass *m*, *k* = 1, 2 marking of chassis 1 and 2, respectively, *i* = 1, 4 belonging to the half-point *A*_111_, respectively, *A*_141_ follows from Figure 2.

Displacement of individual point of body are given by
*Point**Vertical displacement**Stiffness constant*
*B*_011_$w_{011}{=w+y}_{011}\varphi_{x}{+x}_{011}\varphi_{y}-{\;w}_{1\;}+\left(y_{011}{+e}_{y}\right)\varphi_{x1,}$$k_{011}$(21)*B*_014_$w_{014}{=w+y}_{014}\varphi_{x}{+x}_{014}\varphi_{y}-{\;w}_{1\;}-\left(y_{014}{+e}_{y}\right)\varphi_{x1,}$$k_{014}$*B*_021_$w_{021}{=w\;-\;y}_{021}\varphi_{x}{+x}_{021}\varphi_{y}-{\;w}_{2\;}+\left(y_{021}-{\;e}_{y}\right)\varphi_{x2,}$$k_{021}$*B*_024_$w_{024}{=w+y}_{024}\varphi_{x}{+x}_{024}\varphi_{y}-{\;w}_{2\;}-\left(y_{024}-{\;e}_{y}\right)\varphi_{x2.}$$k_{024}$

Potential energy of bogies *m*_1_, *m*_2_ and body *m* is
(22)$$E_{p}=\frac{1}{2}\sum_{j=1}^{2}\sum_{k=1}^{4}\sum_{i=1}^{k_{j}}k_{jki}w_{jki}^{2}+\frac{1}{2}\sum_{j=0}^{}\overset{}{\underset{k=1,2}{\sum }}\sum_{i=1,4}^{}k_{jki}w_{jki}^{2}.$$

Kinetic energy from (18) and potential energy from (22) are substituted into the Equation (4) resulting in
(23)$$\frac{d}{dt}\frac{\partial E_{k}}{\partial {\dot{q}}_{j}}\;-\;\frac{\partial E_{k}}{\partial q_{j}}\;+\;\frac{\partial E_{p}}{\partial q_{j}}\;=\;Q_{j}\;\;\;\;j\;=\;1,\;\ldots ,\;9,$$
where *j*—degree of system freedom, **q***_j_* = (*φ_x_*, *φ_y_*, *w*, *w*_1_, *w*_2_, *φ_x_*_1_, *φ_y_*_1_, *φ_x_*_2_, *φ_y_*_2_)*^T^* and *Q_j_*—generalised forces.

The solution (after derivation with respect to individual coordinates) is given by motion equations in matrix form
(24)$$\begin{array}{c} \left[\begin{array}{ccccccccc} J_{x} & -D_{xy} & 0 & 0 & 0 & 0 & 0 & 0 & 0\\ -D_{xy} & J_{y} & 0 & 0 & 0 & 0 & 0 & 0 & 0\\ 0 & 0 & m & 0 & 0 & 0 & 0 & 0 & 0\\ 0 & 0 & 0 & m_{1} & 0 & 0 & 0 & 0 & 0\\ 0 & 0 & 0 & 0 & m_{2} & 0 & 0 & 0 & 0\\ 0 & 0 & 0 & 0 & 0 & J_{x1} & 0 & 0 & 0\\ 0 & 0 & 0 & 0 & 0 & 0 & J_{y1} & 0 & 0\\ 0 & 0 & 0 & 0 & 0 & 0 & 0 & J_{x2} & 0\\ 0 & 0 & 0 & 0 & 0 & 0 & 0 & 0 & J_{y2} \end{array}\right]\cdot \left[\begin{array}{c} {\ddot{\varphi }}_{x}\\ {\ddot{\varphi }}_{y}\\ \ddot{w}\\ {\ddot{w}}_{1}\\ {\ddot{w}}_{2}\\ {\ddot{\varphi }}_{x2}\\ {\ddot{\varphi }}_{y1}\\ {\ddot{\varphi }}_{x2}\\ {\ddot{\varphi }}_{y2} \end{array}\right]+\\ +\left[\begin{array}{ccccccccc} \kappa_{11} & \kappa_{12} & \kappa_{13} & \kappa_{14} & \kappa_{15} & \kappa_{16} & 0 & \kappa_{18} & 0\\ \kappa_{21} & \kappa_{22} & \kappa_{23} & \kappa_{24} & \kappa_{25} & \kappa_{26} & 0 & \kappa_{28} & 0\\ \kappa_{31} & \kappa_{32} & \kappa_{33} & \kappa_{34} & \kappa_{35} & \kappa_{36} & 0 & \kappa_{38} & 0\\ \kappa_{41} & \kappa_{42} & \kappa_{43} & \kappa_{44} & 0 & \kappa_{46} & \kappa_{47} & 0 & 0\\ \kappa_{51} & \kappa_{52} & \kappa_{53} & 0 & \kappa_{55} & 0 & 0 & \kappa_{58} & \kappa_{59}\\ \kappa_{61} & \kappa_{62} & \kappa_{63} & \kappa_{64} & 0 & \kappa_{66} & \kappa_{67} & 0 & 0\\ 0 & 0 & 0 & \kappa_{74} & 0 & \kappa_{76} & \kappa_{77} & 0 & 0\\ \kappa_{81} & \kappa_{82} & \kappa_{83} & 0 & \kappa_{85} & 0 & 0 & \kappa_{88} & \kappa_{89}\\ 0 & 0 & 0 & 0 & \kappa_{95} & 0 & 0 & \kappa_{98} & \kappa_{99} \end{array}\right]\cdot \left[\begin{array}{c} \varphi_{x}\\ \varphi_{y}\\ w\\ w_{1}\\ w_{2}\\ \varphi_{x2}\\ \varphi_{y1}\\ \varphi_{x2}\\ \varphi_{y2} \end{array}\right]=\left[\begin{array}{c} 0\\ 0\\ 0\\ Q_{4}\\ Q_{5}\\ Q_{6}\\ Q_{7}\\ Q_{8}\\ Q_{9} \end{array}\right]\cdot \end{array}$$

The individual elements of the stiffness matrix *κ_ij_* are functions of stiffness constants *k_jki_*, the dimensions of support *x_jki_*, *y_jki_* and the eccentricities *e_x_* and *e_y_*. The detailed solution of the individual elements is given, e.g., in [22].

The generalised force functions *Q_j_* ((23) and (24)) are again functions of the stiffness constants of the chassis springs *k_jki_* and functions of the road unevenness (elevation) at time *h*(*t*) at the location of the wheel springs. Then the kinematic excitation functions in Equation (24) are given by
(25)$$\begin{array}{c} Q_{1}{=Q}_{2}{=Q}_{3}=0,\\ Q_{4}{=k}_{111}h_{111}{+k}_{121}h_{121}{+k}_{131}h_{131}{+k}_{141}h_{141},\\ Q_{5}{=k}_{211}h_{211\;}{+k}_{221}h_{221}{+k}_{231}h_{231}{+k}_{241}h_{241},\\ Q_{6}=-{\;k}_{111}h_{111}y_{111}-{\;k}_{121}h_{121}y_{121}{+k}_{131}h_{131}y_{131}{+k}_{141}h_{141}y_{141},\\ Q_{7}{=k}_{111}h_{111}x_{111}-{\;k}_{121}h_{121}x_{121}-{\;k}_{131}h_{131}x_{131}{+k}_{141}h_{141}x_{141},\\ Q_{8}=-{\;k}_{211}h_{211}y_{211}{+k}_{221}h_{221}y_{221}-{\;k}_{231}h_{231}y_{231}{+k}_{241}h_{241}y_{241},\\ Q_{9}{=k}_{211}h_{211}x_{211}-{\;k}_{221}h_{221}x_{221}-{\;k}_{231}h_{231}x_{231}{+k}_{241}h_{241}x_{241} \end{array}$$
where *k_jki_h_jki_*(*t*)—the shape of the time function of kinematic excitation at individual axle points and their sequence.

Equation (24) can be rearranged to
(26)$$M_{h}{\ddot{q}}_{j}{+M}_{k}q_{j}{=Q}_{j}\left(t\right).$$

Then the equation is multiplied by matrix **D** (*d_ij_*) from the left, where *d_ij_* = 1/*α_ij_*—diagonal elements of mass matrix **M***_h_*(*α_ij_*). After arrangement, the result is [25,26]
(27)$$M{\ddot{q}}_{j}{+Kq}_{j}{=F}_{j}\left(t\right).$$

Mass matrix can be expressed in form **M** = **E** + **S** and is substituted into (26)
(28)$$\left(E+S\right){\ddot{q}}_{j}{+Kq}_{j}{=Q}_{j}\left(t\right),$$
where elements of matrix **S** are *s_ij_*. These elements are given by equations *s*_11_ = *s*_22_ = 0; *s_ij_* = 0 for *i* = 3, …, 9; *j* = 3, …, 9; *s*_12_ = −*D_xy_*/*J_x_*; *s*_21_ = −*D_xy_*/*J_y_*. These quantities represent the effect of asymmetric distribution of mass, i.e., rotation of main central inertia axes to central axes parallel with geometric axes of the plate (vehicle). Similarly, the elements *a_ij_* of matrix **K** are determined by division of *i*-th row of stiffness matrix by *i*-th element at diagonal of inertia matrix. Excitation function of time *F_j_*(*t*) is determined by the same procedure with function *Q_j_*(*t*).

System of motion equations in case of symmetry is transformed into
(29)$$E{\ddot{q}}_{j}{+Kq}_{j}{=F}_{j}\;\;\;\;\;\mathrm{for}\;i=1\;\ldots \;9.$$
where stiffness matrix is
(30)$$K=\left[\begin{array}{ccccccccc} a_{11} & 0 & 0 & 0 & 0 & a_{16} & 0 & a_{18} & 0\\ 0 & a_{22} & 0 & a_{24} & a_{25} & 0 & 0 & 0 & 0\\ 0 & 0 & a_{33} & a_{34} & a_{35} & 0 & 0 & 0 & 0\\ 0 & a_{42} & a_{43} & a_{44} & 0 & 0 & 0 & 0 & 0\\ 0 & a_{52} & a_{53} & 0 & a_{55} & 0 & 0 & 0 & 0\\ a_{61} & 0 & 0 & 0 & 0 & a_{66} & 0 & 0 & 0\\ 0 & 0 & 0 & 0 & 0 & 0 & a_{77} & 0 & 0\\ a_{81} & 0 & 0 & 0 & 0 & 0 & 0 & a_{88} & 0\\ 0 & 0 & 0 & 0 & 0 & 0 & 0 & 0 & a_{99} \end{array}\right].$$

Elements of stiffness matrix are function of spring stiffness *k*_12_ and *k*_0_ and function of displacement *x*_12_, *y*_12_, *x*_0_, *y*_0_. In the case of complete symmetry, the system of nine differentiated simultaneous equations is transformed to the system of seven simultaneous equations and two independent equations of harmonic motion for *q*_7_ and *q*_9_(*y*_9_).
(31)$${\ddot{q}}_{7}{+a}_{77}q_{7}{=F}_{7}\left(t\right)\;\;\;\;\;{\ddot{y}}_{9}{+a}_{99}y_{9}{=F}_{9}\left(t\right).$$

The solution of the system of Equation (26) can be performed by any of the numerical methods (MATLAB, MAPLE, etc.). Analytical solution is possible by applying matrix calculus, Lagrange’s method of variance of constants, or by Laplace transform. The Laplace transformation yields a system of linear algebraic equations for zero initial conditions [27].

As indicated, the solution can also be done by applying the Laplace transformation, where Equation (27) after modifications becomes
(32)$$[\left(E+S\right)p^{2}+K]{\bar{y}}_{i}(p)\;=\;{\bar{F}}_{i}(p),$$
where ${\bar{y}}_{i}(p)$ and ${\bar{F}}_{i}(p)$ are images of function *q_j_*(*t*) and *F_j_*(*t*) for *j* = 4 … 9, *p*—parameter of Laplace transformation. By rewriting the Equation (32) into the matrix form, it is recommended to solve the system of linear algebraic equations by Cramer rule
(33)$${\bar{y}}_{i}\left(p\right)=\frac{D_{j}\left(p\right)}{D\left(p\right)}=\sum_{I=4}^{n=9}{\left(-1\right)}^{j+i}{\bar{F}}_{i}(p)\frac{D_{ji}\left(p\right)}{D\left(p\right)},$$
where *D*(*p*)—determinant of matrix of above-mentioned system of equations, which is defined as
(34)$${D(p)=C}_{A}{I\;\;\;\;\;\;\mathrm{for}\;n=9\;\;\;\;\;\;C}_{A}=1-{\;s}_{12}s_{21}.$$

Equations for evaluation of real coefficients *A*_2(*n* − *i*)_ can be found in [12]. These equations are generally valid for *n* ≥ 2 and 0 ≤ *i* < *n*.

Equations for evaluation of coefficients *A*_2(*n* − *i*)_ for 2 < *i* < *n* are very complex and should be determined using MAPLE or MATLAB.

To determine the search required function *q_j_*(*t*) by reverse transformation of the images ${\bar{y}}_{i}(p)$, it is convenient to modify the ratio of the determinants *D_ji_*(*p*) and *D*(*p*) so that the product of the expression and image ${\bar{F}}_{i}(p)$ obtained by the modification allows the application of the image convolution theorem.

The polynomial in Equation (34) with real coefficients can be replaced by the product of quadratic binomials
(35)$$\sum_{i=0}^{n}A_{2\left(n\;-\;1\right)}p^{2\left(n\;-\;1\right)}=\prod_{i=1}^{n}\left(p^{2}{+\omega }_{i}^{2}\right).$$

After performing the product on the right side, a polynomial is obtained
(36)$$\sum_{i=0}^{n}A_{2\left(n\;-\;1\right)}p^{2\left(n\;-\;1\right)}=\sum_{i=0}^{n}B_{2\left(n\;-\;i\right)}p^{2\left(n\;-\;i\right)},$$
where real coefficients $B_{2\left(n-i\right)}$ are given by
(37)$$\begin{array}{cc} B_{2n}=1, & B_{2\left(n\;-\;1\right)}=\sum_{i=1}^{n}\omega_{i}^{2},\\ B_{2\left(n\;-\;2\right)}=\sum_{i=1}^{n\;-\;1}\sum_{j=i+1}^{n}\omega_{i}^{2}\omega_{j}^{2}, & B_{2\left(n\;-\;3\right)}=\sum_{i=1}^{n\;-\;2}\sum_{j=i+1}^{n\;-\;1}\sum_{l=j+1}^{n}\omega_{i}^{2}\omega_{j}^{2}\omega_{l}^{2},\\ B_{2\left(n\;-\;4\right)}=\sum_{i=1}^{n\;-\;3}\sum_{j=l+1}^{n\;-\;2}\sum_{j=1}^{n\;-\;1}\sum_{k=l+1}^{n}\omega_{i}^{2}\omega_{j}^{2}\omega_{l}^{2}\omega_{k}^{2}, & B_{2\left(n\;-\;n\right)}{=B}_{0}\prod_{i=1}^{n}\omega_{i}^{2}. \end{array}$$

By comparing the coefficients of the polynomials for the same powers of the parameter *p^2(n − i)^* on both sides of Equation (36) *A*_2(*n* − *i*)_ = *B*_2(*n* − *i*)_ a system of equations is obtained to determine *ω_i_*, *ω_j_*, *ω_l_*, ….

By comparing Equation (37) with the relations determining the coupling of the coefficients of the algebraic equation and its root factors, the algebraic equation is obtained
(38)$$\begin{array}{c} f\left(\omega^{2}\right){=\omega }^{2n}-{\;A}_{2\left(n-1\right)}\omega^{2\left(n\;-\;1\right)}{+A}_{2\left(n\;-\;2\right)}\omega^{2\left(n\;-\;2\right)}-{\;A}_{2\left(n\;-\;3\right)}\omega^{2\left(n\;-\;3\right)}+\\ +{\left(-1\right)}^{n\;-\;1}A_{2}\omega^{2\;}+{\left(-1\right)}^{n}A_{0}=0. \end{array}$$

Determining the roots of the frequency Equation (38) is the most complicated part of the proposed solution, especially in terms of numerical accuracy, i.e., *ω*^2^—the circular frequencies of the functions *y_j_*(*t*), for *j* = 1, 2, 3, …, *n*. The next solution, after modifications and back-transformation, would give the convolution integral
(39)$$q_{j}\left(t\right)=\sum_{i=4}^{n}{\left(-1\right)}^{j+i}\sum_{k=1}^{n}\frac{L_{ji,k}}{\omega_{k}}\int_{0}^{t}F_{i}\left(\tau \right){\sin\omega }_{k}\left(t\;-\;\tau \right)d\tau$$
where *ω_k_* is the solution of Equation (38). From the known functions *q_j_*(*t*), i.e., the solution of the system of Equation (26), the required quantities of the system of Equation (24) are determined.

This method makes it possible to determine the vertical displacements of arbitrary points of the chassis or body of the vehicle, for example, the displacements of the points described by Equations (19)–(21). Thus, a program can be developed to calculate the vertical displacements of arbitrary points of railway vehicle chassis models with a large range of design variations.

## 3. Experimental Methods

The above method was experimentally verified on a real railway vehicle. A four-axle railway freight car of the Smmps series was used for the experiment. The aim of the measurements was to determine the geometrical, mass, and stiffness parameters of the single-individual parts of the freight car. The main objective was to establish the oscillations of the individual parts of the wagon when crossing the wedges at different wedge locations and load placements on the wagon platform (wedges = kinematic excitation, load placement = symmetrical or asymmetrical load). The measurement requirements were specified, and the wagon was modified for these specific requirements.

### 3.1. Description of Vehicle Arrangement

Due to the requirements for suspension linearity, the original chassis with leaf springs (type 26-2.8) were replaced by chassis with coil springs (Y25). For the new chassis, the outer springs of the primary suspension were removed (Figure 3a), and a secondary suspension was created by placing three springs on the longitudinal beam of each chassis (Figure 3b).

The bogies were fixed to the body (car platform) by a network of steel cables with cable tensioners—Figure 4a.

For all springs (internal and external) of both chassis, their characteristics and stiffnesses were determined, and based on these characteristics the selection was made so that the resulting stiffnesses of the primary and secondary suspension on the wheel, wheelset, chassis, and car were distributed with maximum symmetry to the longitudinal and transverse axes of the car.

The flatcar was loaded with two loading units (Figure 4b), each weighing 4000 kg (the units were made up of concrete blocks with dimensions of 4900 × 1000 × 300 mm). For the chassis, the masses of the parts, their centre of gravity coordinates, and the moments of inertia to the axes passing through the centre of gravity were experimentally determined. Similar parameters were determined for the vehicle body. For these measurements, a proprietary methodology was developed.

### 3.2. Methodology

The modified vehicle was subjected to wedge crossing tests (Figure 5). The car was loaded with loading units (either symmetrically or unsymmetrically). The location of the wedges was varied during the tests depending on the kinematic excitation requirement (symmetrical, asymmetrical, single axle, both axles, single chassis, both chassis).

The relative vertical displacement of the chassis frame relative to the wheelset (9 sensors), the car body relative to the chassis frame (4 sensors), and the vertical acceleration of selected points on the body (5 sensors) and chassis (2 sensors) were measured. The location of the sensors can be seen in Figure 6. During the tests, time histories of the measured quantities were recorded and used to verify the theoretical model.

Strain gauge track sensors (chassis frame × wheelset) and cable track sensors (box frame × chassis frame) were used to measure vertical displacement. Acceleration was measured with B12/200 Hz inductive sensors (Figure 7).

The experiments were carried out according to an optimised test plan. Selected combinations of experiments arose from statistical analysis of combinations of all possible experimental settings. These combinations were sufficient to achieve the goals, i.e., to identify critical situations in which derailment may occur [28]. The tests were carried out for different cases of load distribution on the car platform and wedge placement on the rails, taking into account the wheelbase of the bogies (1800 mm) and the car wheelbase (9400 mm). The test plan can be seen in Figure 8, where the different variants of the car crossing the wedges are shown. A total of 30 combinations were measured. Each measurement was repeated 3 times and a total of 90 tests were performed. Prior to the actual car crossings over the wedges, the front and rear heights of the bogie frames and car body above the tops of both rails were measured. The static seating of the primary and secondary suspension springs under a given load (symmetrical, unsymmetrical) was checked. The car was pulled over the wedges by a motorised locomotion system using a steel cable. The wedges were attached to the rail sections that were part of the block for dynamic measurements.

### 3.3. Experimental Results

Due to the scope of the measurements, only selected results for symmetrical and asymmetrical loading and for symmetrical and asymmetrical excitation are presented in this paper. The labels of the locations where the relative deflections between the case and chassis, chassis, and wheelset were measured are shown in Figure 9.

(a)The time trend of the deflections when the car crosses the wedges under symmetrical loading of the car and symmetrical kinematic excitation is shown in Figure 10 and Figure 11.

In this case, the symmetrical load is represented by the symmetrical positioning of the load with respect to the geometric centre of the car. Figure 10 shows the time history of the vertical displacement of the first wheelset relative to the chassis frame. The waveform is symmetric about zero displacement. The only damping that affects the deflection is material damping. The variation in the vertical displacement is due to slight differences in the material properties of the springs of the first wheelset. The displacements show an excellent agreement.

Figure 11 shows the vertical displacement of the chassis frame at two locations, see Figure 9 (2L, 2R), relative to the car body. Both places are located off the geometric centre of the chassis, therefore, in agreement with Equation (1), the waveforms are displaced off the axis of symmetry. Again, the waveforms show a very good agreement, and it can be seen that the chassis frame is rotated relative to the longitudinal and transverse axes when crossing the wedges.

(b)Time trends of displacement when the car crosses the wedges under symmetrical loading, asymmetrical kinematic excitation—Figure 12 and Figure 13.

Figure 12 shows the vertical displacement between the first wheelset and the chassis frame under symmetrical loading and asymmetrical kinematic excitation (variant A–IV). In this kinematic excitation, there is a different excitation of the first wheelset on the left and right side, which can be seen in the deflection of waveforms. In this kinematic excitation, the wheelsets rotate about the longitudinal axis of the car. The whole chassis and body of the car are also rotated about the longitudinal and transverse axes. The figure also shows that at certain stages the chassis wheel loses contact with the rail (loss of adhesion).

From the above time trend (Figure 13), it is clear that due to the oscillation and the resulting rotation of both the wheelset and the chassis frame at the measuring points, there is also a rotation of the body. This kinematic excitation causes the wheel to bounce significantly off the rail and, under certain conditions (e.g., in a horizontal curve, on a turnout), could cause a derailment due to the large displacement.

(c)Time trends of displacement when the car crosses the wedges under asymmetrical loading, symmetrical kinematic excitation—Figure 14 and Figure 15.

The displacement of the front wheelset relative to the chassis frame (Figure 14) show a difference in position relative to the zero deflection axis even under symmetrical kinematic excitation and asymmetrical loading. This is due to the non-symmetrical load distribution and hence the rotation of the car body and the different loading on the left and right side of the chassis. The maximum displacement is less than that of variant A-IV.

The relative deflection between the chassis frame and the car body is again shifted relative to the axis through the centre of gravity in the case D-I, i.e., under asymmetrical loading and symmetrical kinematic excitation (Figure 15). The displacements are much larger at their maximum values than between the chassis frame and the wheelset. The difference in their trends is due to the fact that in one case (2R) the displacement is in the axis of transverse symmetry of the chassis frame. In 2L case the displacement is affected by rotation and displacement from the transverse axis—see Equation (1). The maximum deflections are again quite significant and reach a maximum of about 18 mm.

(d)Time trends of displacement when the car crosses the wedges under asymmetrical loading, asymmetrical kinematic excitation—Figure 16 and Figure 17.

The load distribution is the same as in the previous case, but the kinematic excitation is asymmetrical (Figure 16). Displacement trend between the chassis frame and the wheelset reaches more than 20 mm at the maximum (this is less than the wedge height). The decay to an acceptable value occurs relatively quickly, in about 20 s. It is also clearly visible in the figure that the first and second wheelsets are repeatedly dropped on each side (the wedges are directly connected to each other).

The difference in the position of the vertical displacement between the car body and the chassis frame (Figure 17) at points 2L and 2R is due to the different position of these points relative to the transverse axis of the chassis frame and the transverse axis of the car body. The magnitude of the displacement is influenced by the asymmetrical loading (variant D-IV) which caused the car body to rotate relative to its longitudinal and transverse axes. The consequence is also a different load on the left and right side of all wheelsets.

In addition to the measurements of the relative vertical displacement, measurements of the vertical acceleration were also carried out. Most of the acceleration measurement points were located on the vehicle body (five sensors in total). In this paper we present the vertical acceleration waveform of the vehicle body in its transverse axis on the left and right side (points 3L and 3R) for asymmetrical loading and asymmetrical kinematic excitation (variant B-III). To illustrate this situation, the time history of the relative displacements at points 1L, 1R (Figure 18), 2L, 2R (Figure 19) is also shown for this case.

Figure 18 shows the relative displacement between the chassis frame and the wheelset (B-III). This variant is characterised by an asymmetrical load (load placed on the right side of the car body symmetrical to the transverse axis). The kinematic excitation is also asymmetrical. This load has caused the car body to rotate about the longitudinal axis and thus also the load on the chassis to be unequal. From the displacement waveforms it is evident that the larger amplitude after dropping from the wedges are on the unloaded side. After the last wedge jump on the loaded side the displacement is maximum. The displacement on the unloaded side reached almost the same values immediately after the wedge crossings. A gradual damping of the deflections occurs after 27 s. On the unloaded side, there is a significant loss of wheel-rail contact after each wedge, while on the loaded side there are only two significant peaks, after the first and last wedges.

The relative displacement between the car body and the chassis frame (Figure 19) are relatively far apart (about 35 mm) due to the rotation of the car body and therefore the different loads on the left and right sides. The time histories of the deflections are almost identical, but it is clear that the left side was already lightened before the test. The maximum deflections of the loaded side are higher than those of the unloaded side when jumping off the wedges, which is consistent with the nature of the oscillation of the car body. Such a load distribution under asymmetrical kinematic excitation may also cause the car to derail.

The vertical acceleration in the middle of the car body on both the left and right sides (Figure 20) was higher on the unloaded side (3L) than on the loaded side (3R) at the peak of the wedge jump. The maximum value was reached after crossing the first wedge, and the maxima decreased when crossing the other wedges.

## 4. Conclusions

The paper deals with the theory of oscillations of mechanical systems of elastically supported and bound bodies. The application of this solution to vehicle models has been carried out. The methodology of the model solution is proposed and the application of this methodology to a real vehicle under different operating conditions has been carried out. The application has been made, among other things, to a railway chassis vehicle—its solution is the subject of this paper. Based on the developed methodology, an experimental solution of a four-axle railway vehicle without damping (shock absorbers) was implemented. The results of the experimental solution, i.e., the time course of vertical displacement, are presented in the previous section.

From the given results, it can be seen that in the case of asymmetrical kinematic excitation (e.g., rail fracture, unevenness on the rail), the vertical oscillation of the car will cause the vertical oscillation. The consequence of this oscillation is the loss of contact between the wheel and the rail (loss of adhesion). In any case where the critical value of vertical displacement of the wheel, which is equal or higher than height of axle, is achieved, the vehicle may be derailed. These vertical deflections are also strongly influenced by the asymmetry of the weight distribution of the vehicle and, in particular, by the asymmetry of the load. From the results obtained, it is necessary to pay particular attention to the distribution of the load (especially piece loads) on the railway carriage.

## Figures and Tables

**Figure 1 sensors-22-04033-f001:**
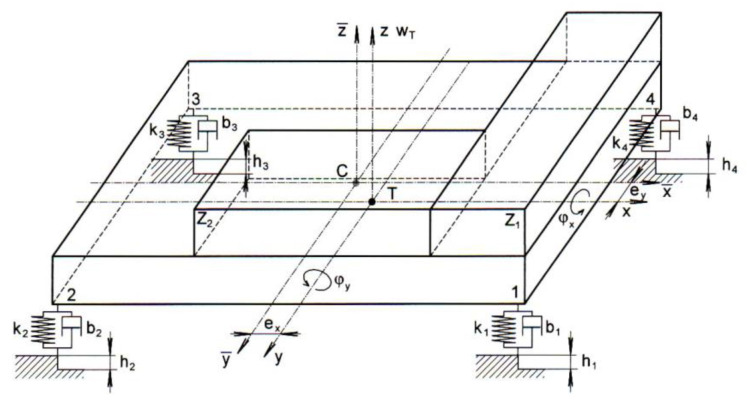
Simple general model of the vehicle.

**Figure 2 sensors-22-04033-f002:**
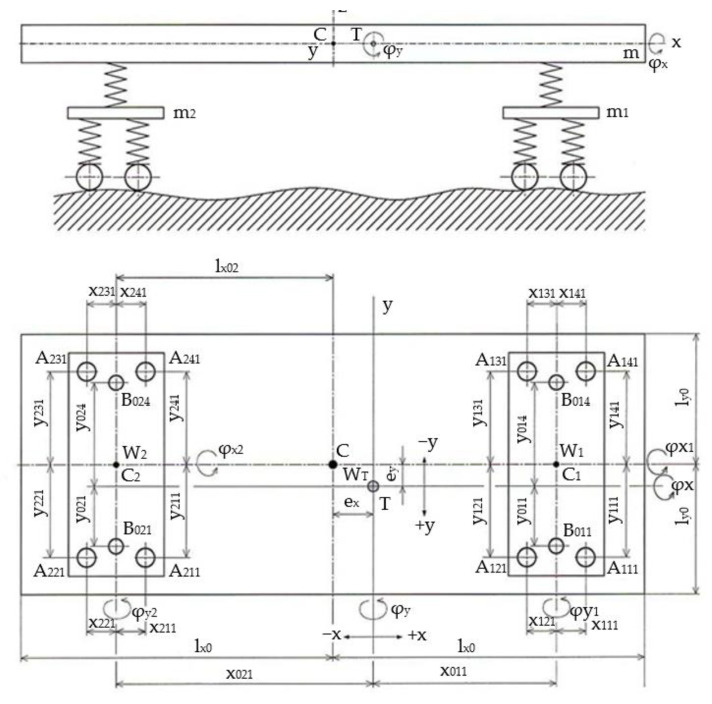
The scheme of the model.

**Figure 3 sensors-22-04033-f003:**
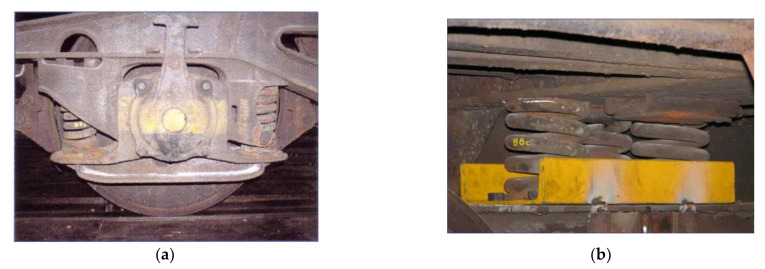
Chassis support: (**a**) Original springs; (**b**) substituted springs.

**Figure 4 sensors-22-04033-f004:**
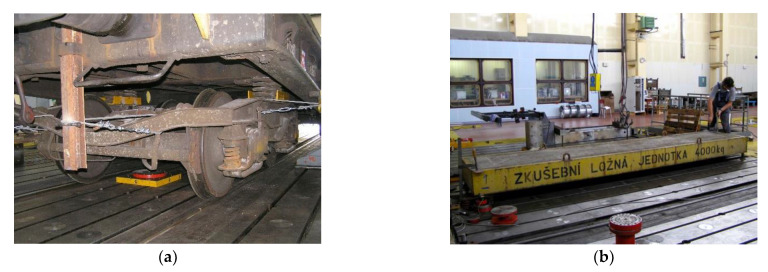
(**a**) Bogie fixed to the car platform; (**b**) loading unit.

**Figure 5 sensors-22-04033-f005:**
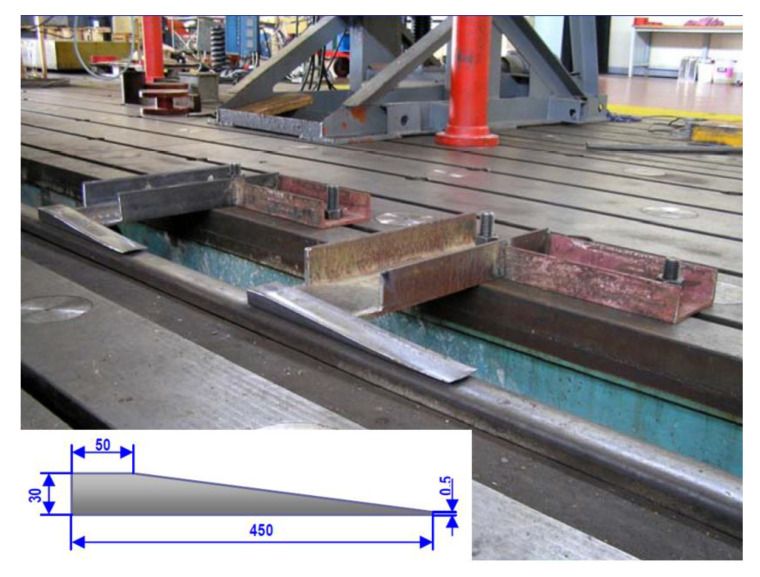
Wedge locations at rail.

**Figure 6 sensors-22-04033-f006:**
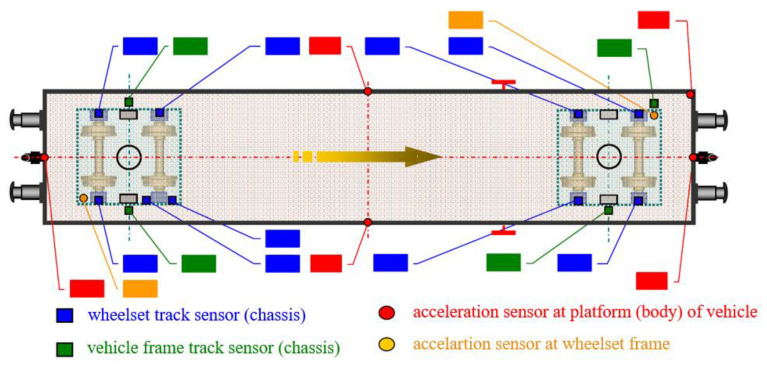
Sensor locations.

**Figure 7 sensors-22-04033-f007:**
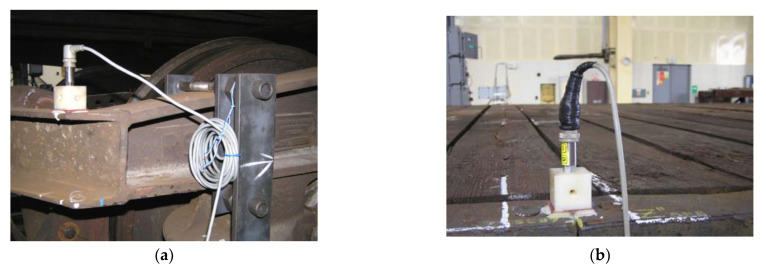
Acceleration sensor location: (**a**) at frame of backward car; (**b**) at platform of car—transversal axis.

**Figure 8 sensors-22-04033-f008:**
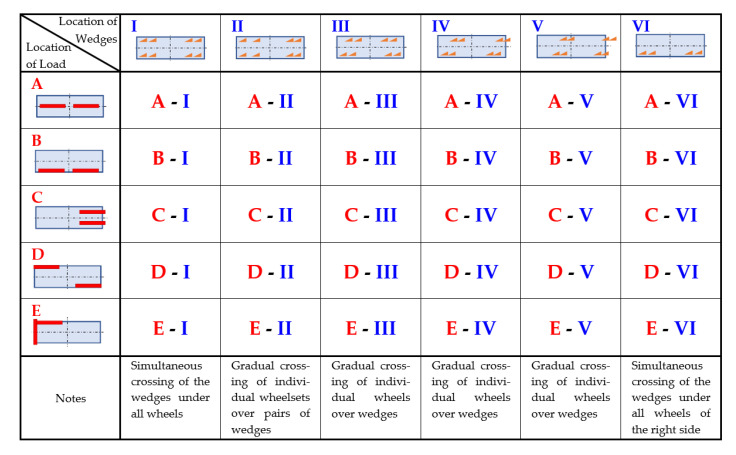
Measured combination of loading and kinematic excitation.

**Figure 9 sensors-22-04033-f009:**
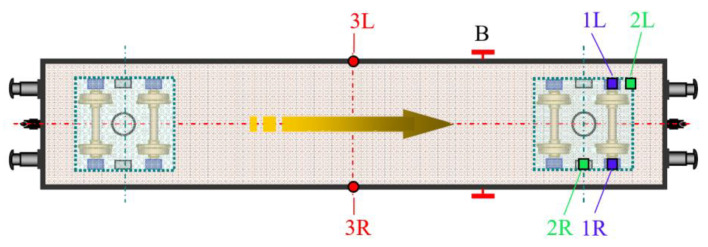
Location of relative vertical deflection and vertical acceleration sensors at selected points. B—handbrake, 1L/R—chassis frame vertical deflection sensor, first wheelset, left/right side, 2L/R—chassis frame and body vertical deflection sensor, left/right side, 3L/R—car body vertical acceleration sensor, left/right side.

**Figure 10 sensors-22-04033-f010:**
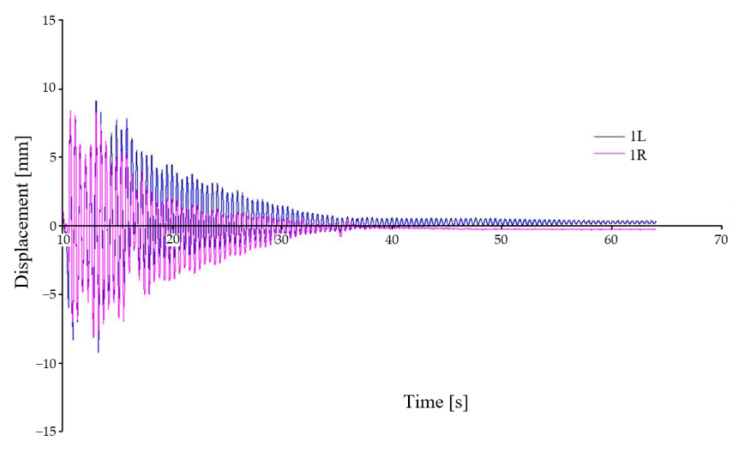
Relative vertical displacement between chassis frame and wheelset—variant A–I.

**Figure 11 sensors-22-04033-f011:**
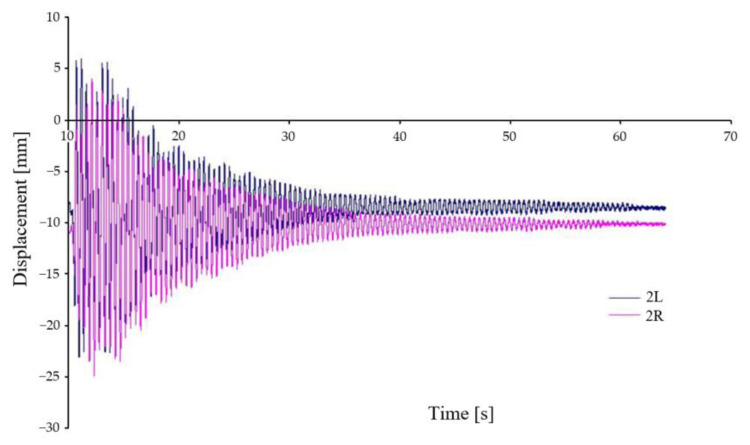
Relative displacement between car body and chassis frame—variant A–I.

**Figure 12 sensors-22-04033-f012:**
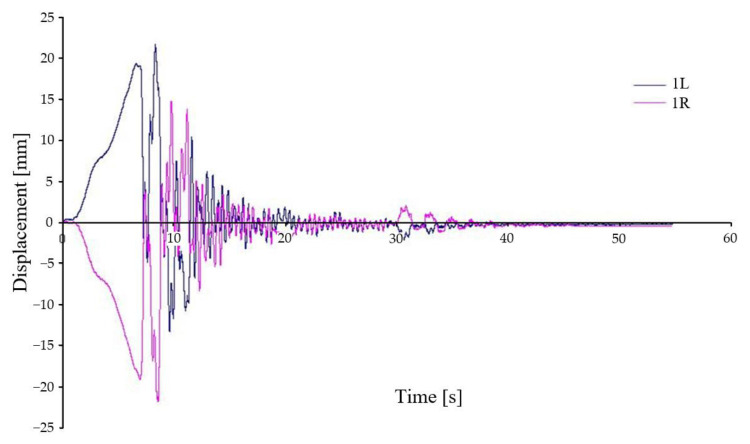
Relative displacement between chassis frame and wheelset—variant A–IV.

**Figure 13 sensors-22-04033-f013:**
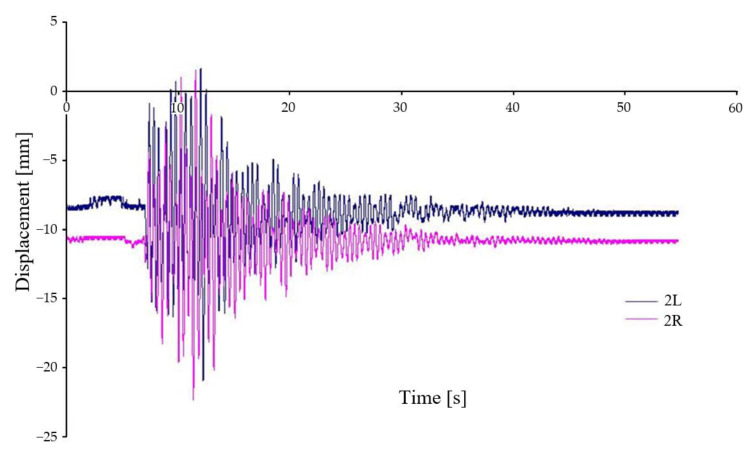
Relative displacement between car body and chassis frame—variant A-IV.

**Figure 14 sensors-22-04033-f014:**
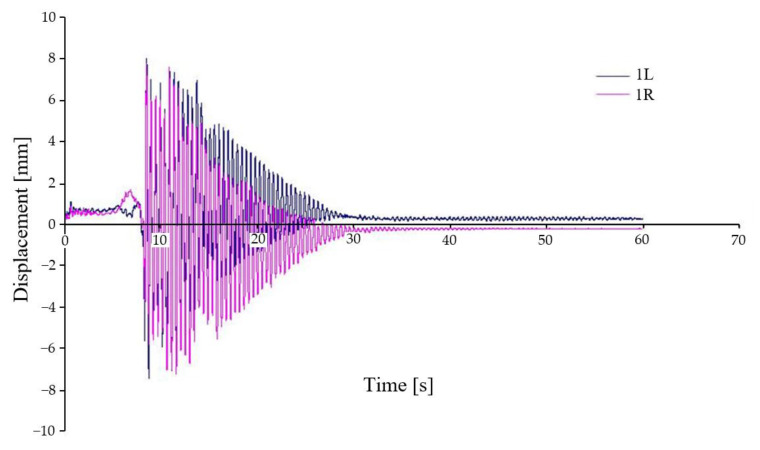
Relative displacement between chassis frame and wheelset—variant D-I.

**Figure 15 sensors-22-04033-f015:**
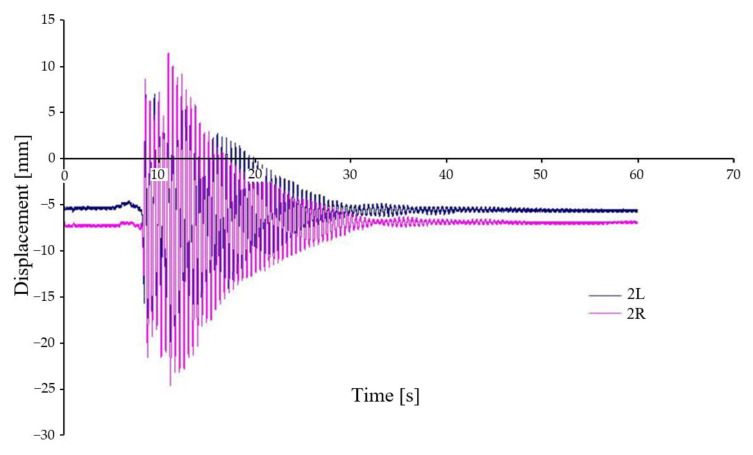
Relative displacement between car body and chassis frame—variant D-I.

**Figure 16 sensors-22-04033-f016:**
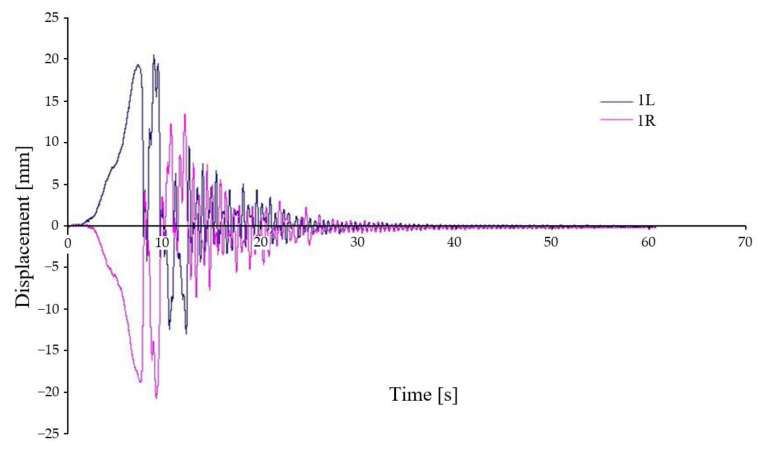
Relative displacement between chassis frame and wheelset—variant D-IV.

**Figure 17 sensors-22-04033-f017:**
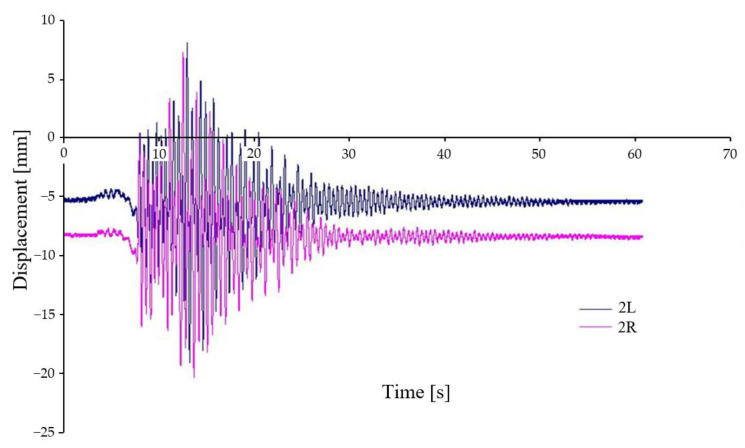
Relative displacement between car body and chassis frame—variant D-IV.

**Figure 18 sensors-22-04033-f018:**
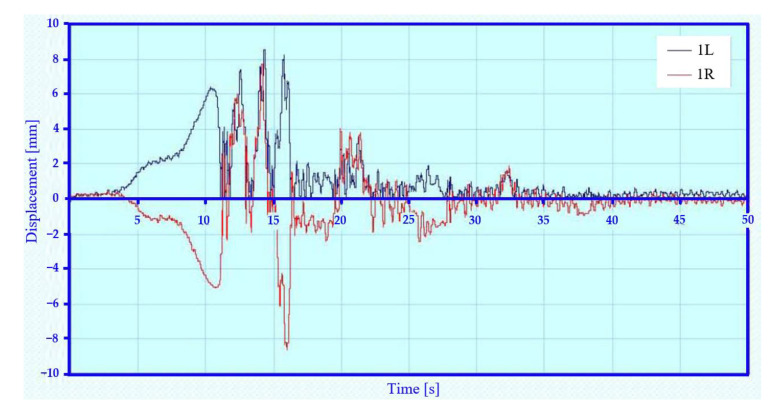
Relative displacement between chassis frame and wheelset—variant B-III.

**Figure 19 sensors-22-04033-f019:**
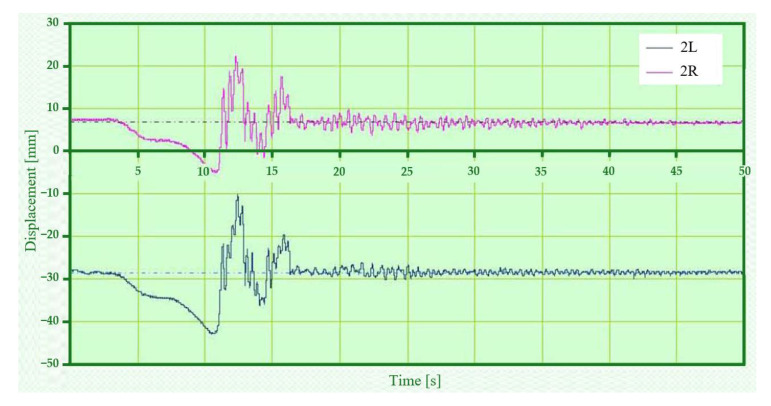
Relative deflections between car body and chassis frame—variant B-III.

**Figure 20 sensors-22-04033-f020:**
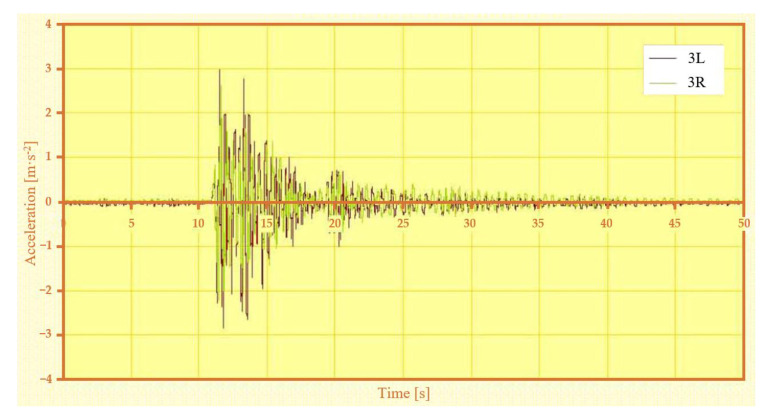
Vertical acceleration in the middle of the car body on the left and right side—variant B-III.

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
