# Peer review of "Vertical Oscillation of Railway Vehicle Chassis with Asymmetry Effect Consideration"

_sensors, 2022, doi:10.3390/s22114033_

Round 1
Reviewer 1 Report
Generally, the reviewer thinks that this is an interesting work to perform the analysis of vertical oscillation of vehicle chassis considering the asymmetry effect. Both the analytical and experimental results are presented, and this work may have a good contribution to the research field. But some editorial and technical issues can be found in the current version. Please sort out or clarify them before the publication.
Editorial issues:
The necessity of studying the wheel-rail interaction is recommended to be indicated with a bit literature review. It is of great importance to safety and comfortability [1], and also the vehicle-overhead line interaction [2].
[1] Zhai W. Vehicle–Track Coupled Dynamics. Veh. Coupled Dyn. Springer Singapore; 2020.
[2] Song Y, Wang Z, Liu Z, et al. A spatial coupling model to study dynamic performance of pantograph-catenary with vehicle-track excitation. Mech Syst Signal Process. 2021;151:107336.
The subsection titles are very confusing. 2.1.1 spatial model, and 2.1.2 spatial model. Please clarify and modify.
The title of section 2.2 is too long. A more specific one is desired.
Subsection 3.3 is the analysis of the results. But the title of section 3 is ‘experimental methods and discussion’. It sounds like the introduction of methods and discussion of the differences. Please modify to avoid misunderstanding.
Technical issues:
- The most interesting part would be the comparison between the analytical results and the experimental results. But the reviewer does not find this information. Can the authors do validation before the analysis?
- At the beginning of section 2, the authors introduced ‘elastically supported rigid body’. But in Section 2.1.1, why is a flexible model presented? Please explain the contradiction description.
- No comparative analysis is presented in Section 2.1.4. How can it be ‘model comparison’?
Author Response
Dear Editor and Reviewers,
Firstly, all authors would like to thank you for all the comments and suggestions of the
reviewers. We are grateful to the editor of the Journal for giving us the opportunity to submit
the manuscript. We believe the changes are complete and are in accordance with the
recommendations of the reviewers and the editor. Please find our responses to the reviewers’
comments and questions below. Major substantive or technical changes are marked up using
the “Track Changes” function in the manuscript.
Frantisek Klimenda

Reviewer 2 Report
The introduction is less state-of-art and the construction of the content looks like a technical report.
To assess the derailment safety, the wheel unloading ratio and Nadal L/V ratio are commonly used as well as the wheel lift displacement. It is not scientific to assess the derailment safety through the suspension defection without scientific proof.
- Concerning the wedge tests, why only six excitation cases were concerned? As demonstrated in the following reference, the pitch, roll and lateral excitations were also measured to identify the damping ratio of the vehicle system. Shi H, Wu P, Luo R and Zeng J. Estimation of the Damping Effects of Suspension Systems on Railway Vehicles Using Wedge Tests[J]. Proceedings of the Institution of Mechanical Engineers, Part F: Journal of Rail and Rapid Transit, 2016, 230(2): 392-406.
- How the experimental results, the time history of displacements and accelerations, are used to identify the vehicle mass eccentricity? Shall any limit be set for the running safety concern?
- The model section is separated from the experimental section. In other words, why there is no simulation to explain the test results?
- In the conclusion section, especially in the second paragraph, too many ‘if’ and ‘may’ rephrases are used without any solid proof declared in the content.
Author Response

(The authors gave the same response as above.)

Round 2
Reviewer 1 Report
I recommend the publication of this paper with the current version.
Author Response
Dear Reviewer, thank you for your effort spent to revision of our article.
Reviewer 2 Report
Regarding the first comment about the excitation cases, we see the authors have done lots of tests but the comment is about why they are set like that? For the vertical excitation on the vehicle, except for exciting the eight wheels at the same time, it is possible to excite only the front or the rear bogie. Please give a discussion in the content about the test set by referring to the suggested reference titled wedge tests.
Author Response

(The authors gave the same response as above.)
